# Preparation of Spherical Cellulose Nanocrystals from Microcrystalline Cellulose by Mixed Acid Hydrolysis with Different Pretreatment Routes

**DOI:** 10.3390/ijms231810764

**Published:** 2022-09-15

**Authors:** Peng Zhu, Luyao Feng, Zejun Ding, Xuechun Bai

**Affiliations:** College of Textile Science and Engineering (International Institute of Silk), Zhejiang Sci-Tech University, Hangzhou 310018, China

**Keywords:** spherical cellulose nanocrystals, pretreatment, microcrystalline cellulose, mixed acid hydrolysis, crystallinity, thermal stability

## Abstract

Spherical cellulose nanocrystal (CNC), as a high value cellulose derivative, shows an excellent application potential in biomedicine, food packaging, energy storage, and many other fields due to its special structure. CNC is usually prepared by the mixed acid hydrolysis method from numerous cellulose raw materials. However, the pretreatment route in preparing spherical CNC from cellulose fiber is still used when choosing microcrystalline cellulose (MCC) as the raw material, which is not rigorous and economical. In this work, pretreatment effects on the properties of spherical CNC produced from MCC by mixed acid hydrolysis were systematically studied. Firstly, the necessity of the swelling process in pretreatment was examined. Secondly, the form effects of pretreated MCC (slurry or powder form) before acid hydrolysis in the preparation of spherical CNC were carefully investigated. The results show that the swelling process is not indispensable. Furthermore, the form of pretreated MCC also has a certain influence on the morphology, crystallinity, and thermal stability of spherical CNC. Thus, spherical CNC with different properties can be economically prepared from MCC by selecting different pretreatment routes through mixed acid hydrolysis.

## 1. Introduction

In pursuit of environmentally friendly and sustainable development, the preparation and application of biomass materials have attracted great attention [1]. Cellulose is considered to be the most abundant, biodegradable, renewable, and sustainable biopolymer on Earth and its biosynthesis, chemistry, and ultrastructure remains an active field of research [2,3,4]. As one of the most important derivatives of cellulose, nanocellulose has great potential in biomedicine, food packaging, energy storage, environmental remediation, and many other fields [5]. Cellulose nanofibril (CNF) and CNC are the two main forms of nanocellulose [6,7,8]. CNF (≥1 μm of length) has a larger aspect ratio than CNC, which can form a unique entangled network structure. CNC (about 100–200 nm of length) is much shorter than CNF, while it has a higher crystallinity and a nanometer size. Moreover, CNC has the advantages of great mechanical properties, good biocompatibility, a high specific surface area, biodegradability, and a low thermal expansion coefficient [9,10].

In the past decades, various approaches have been proposed for CNC preparation. The size, shape, morphology, and structural properties of the cellulose nanomaterials were determined by these methods [11]. The morphology of a cellulose nanomaterial is closely related to its application characteristics, which will have a great impact on its application prospects. According to the existing literature, Ding et al. first prepared spherical CNC by a mixed acid hydrolysis method (sulfuric acid/hydrochloric acid) from short-staple cotton in 2000 [12]. As one of the nanocelluloses, spherical CNC has a large specific surface area, uniform particle size, higher bulk density, and higher thermal stability [1]. It has more prominent advantages in many aspects, which has attracted the attention of many researchers. Yu et al. changed the type of mixed acid and also prepared spherical CNC from Lyocell fiber using formic acid/hydrochloric acid, as well as acetic acid/hydrochloric acid [13,14,15]. Furthermore, enzymatic hydrolysis is also a common method to prepare spherical CNC from cotton fiber [16], pulp fiber [1,5,17,18], or bacterial cellulose [19]. Moreover, Cheng et al. prepared spherical CNC through the chemical hydrolysis of Lyocell fiber in an ammonium persulfate solution [20]. In addition, a single acid combined with mechanical action was used to prepare spherical CNC from oil palm empty fruit bunch pulp [21] or sesame husk [22]. Meanwhile, other approaches were also proposed to prepare spherical CNC [2,23,24].

Among the above methods, the mixed acid hydrolysis method (sulfuric acid and hydrochloric acid) is a widely used method to prepare spherical CNC. In these studies, the pretreatment and hydrolysis processes are basically the same as shown in Figure 1a [3,12,25]. Even when the raw material was replaced by commercial MCC, the pretreatment process used was still the same as that of cellulose fibers (Figure 1b) [4]. The sizes of MCC and cellulose fibers are not in the same order of magnitude, and their chemical composition and physical structure are distinctly different [26,27,28]. Moreover, some series of MCC are obtained by acid hydrolysis from cellulose fibers through many high energy consuming steps [29,30,31]. Therefore, using the same pretreatment process from different raw materials is not rigorous and economical, although spherical CNC can be prepared.

In the pretreatment process for cellulose fibers, sodium hydroxide is chosen to hydrolyze the amorphous region and swell fibers at the same time [3,16,22]. Meanwhile, the crystal type can be changed during the sodium hydroxide treatment from a cellulose I to a cellulose II structure, from which it is easier to fabricate spherical CNC [15,24]. DMSO is mainly used to swell cellulose fibers [3,12]. At present, the necessity of the DMSO swelling process in the preparation of spherical CNC by the mixed acid hydrolysis method from MCC requires investigation.

Furthermore, the obtained substance in slurry form after the DMSO swelling process will be hydrolyzed with mixed acid [3,4]. On the one hand, the form of slurry is conducive to maintaining the swelling effect, making it easier for mixed acid to enter the amorphous region or even the crystalline region of cellulose fibers [21]. On the other hand, the water content in the slurry cannot be easily controlled, resulting in poor repeatability, poor long-term storage stability, and excessive acid needed to ensure the product quality.

In the present work, commercial MCC was used as the raw material to prepare spherical CNC by mixed acid hydrolysis. The necessity of the swelling process in pretreatment during the preparation of spherical CNC from commercial MCC was examined. In addition, the form effects of pretreated MCC (slurry or powder form) before mixed acid hydrolysis on the preparation of spherical CNC were carefully investigated. This work will provide support for selecting the pretreatment routes of MCC to prepare desired spherical CNC for further studies.

## 2. Results and Discussion

### 2.1. Chemical Structure

FTIR spectroscopy was used to evaluate the chemical structure of MCC before and after the pretreatment processes. Figure 2a illustrates FTIR spectra of MCC, MCC after alkaline treatment (MCC+NaOH), and MCC after alkaline treatment and swelling treatment by DMSO (MCC+NaOH+DMSO). It can be observed that all samples have the common peaks representative of cellulose: O–H stretching vibrations and flexural vibrations of intra- and intermolecular hydrogen bonds near 3000–3700 cm^−1^ [16,19], C–H stretching vibrations at 2900 cm^−1^ [21], O–H bending of absorbed water at 1647 cm^−1^ [16], H–C–H and O–C–H in-plane bending vibrations at 1430 cm^−1^ [4], C–H deformation vibrations at 1372 cm^−1^ [32], C–O stretching vibrations at 1160 cm^−1^ [22], O–H bending in β-glycosidic linkages between the anhydroglucose units at 897 cm^−1^ [21,33], and C–OH out-of-plane bending at 667 cm^−1^ are characteristic peaks of cellulose [4,32].

The main differences between the FTIR spectra in Figure 2a are the crystalline I characteristic peaks presented in the commercial MCC, O–H stretching vibrations at 3272 cm^−1^, and O–H out-of-plane bending at 710 cm^−1^ [34]. Furthermore, the 1430 and 1111 cm^−1^ absorption peaks can also be used to study the type of crystalline cell and the crystallinity changes because the crystalline cellulose Ⅰ spectra differs clearly in these peaks from cellulose Ⅱ [35]. Ring asymmetric stretching at 1111 cm^−1^ is a characteristic peak of cellulose Ⅰ. Meanwhile, the peak near 1430 cm^−1^ is attributed to H–C–H symmetric bending or a scissoring motion which is known to be the sum of the cellulose Ⅰ peak at 1430 cm^−1^ and the cellulose II peak at 1420 cm^−1^ [36]. Comparing the FTIR spectra of MCC before and after alkaline treatment, it can be seen that the intensity of peak at 1430 cm^−1^ is significantly reduced and characteristic peaks at 3272 cm^−1^, 1111 cm^−1^, and 710 cm^−1^ are absent, indicating the complete conversion of cellulose Ⅰ to cellulose Ⅱ after alkaline treatment. In addition, the peak at 809 cm^−1^ in the commercial MCC spectra is due to glucomannan from hemicellulose [37], which disappeared in the sample spectra after alkaline treatment, meaning that the residual hemicellulose is completely removed during alkaline treatment.

Based on the FTIR analysis from Figure 2a, the molecular structures of pretreated MCC were changed from cellulose Ⅰ to cellulose Ⅱ through the alkaline treatment. As shown in Figure 2b, all CNC samples prepared with different pretreatments maintain the cellulose Ⅱ structure, due to the absence of characteristic peaks of cellulose I at 3272 cm^−1^, 1111 cm^−1^, and 710 cm^−1^.

### 2.2. Micromorphology

Figure 3 demonstrates the FE-SEM images of MCC before and after different pretreatment routes. Compared with original MCC (Figure 3a), MCC+NaOH is smoother and more fragmented, as shown in Figure 3b. Due to the alkaline hydrolysis of cellulose, the surface and some amorphous regions of MCC are hydrolyzed first, leading to the above morphological structure. With further swelling by DMSO (Figure 3c), MCC+NaOH+DMSO becomes much more fragmented as some small organic molecules are dissolved in DMSO. In addition, their morphology becomes more irregular because of the deformation during the drying process from the swelling state.

FE-SEM images of CNC samples prepared with different pretreatments are presented in Figure 4. It can be seen that all CNC samples show an approximately spherical appearance and exhibit a very small size with a narrow size distribution, which has been annotated in Figure 4. In particular, the roundness of CNC(N_P_-D_S_-H) and CNC(N_S_-H) is improved, which is attributed to the easier entry of mixed acid into MCC when MCC is in slurry form. In Figure 4c,d, some nanoparticles with larger sizes are actually aggregates of small spherical CNC, which are easily formed during the natural evaporation of water in the sample preparation process for FE-SEM observation. Through the analysis and calculation of ImageJ software, the average diameters of CNC(N_P_-D_P_-H) and CNC(N_P_-D_S_-H) are 10.4 ± 3.7 nm and 10.1 ± 1.5 nm, respectively. Meanwhile, without any additional swelling treatment, CNC(N_P_-H) and CNC(N_S_-H) display an average diameter of 11.2 ± 1.9 nm and 9.2 ± 2.0 nm, respectively. It can be seen that the average particle size of CNC(N_P_-D_S_-H) and CNC(N_S_-H) is a bit smaller because the MCC has swollen, which makes it easier for the mixed acid solution to enter the MCC, resulting in a higher hydrolysis efficiency. It should be noted that in the preparation of CNC samples without any additional swelling treatment, MCC in slurry form has been initially swelled in the alkaline treatment [3,16,22].

These FE-SEM images strongly demonstrate that spherical CNC can be appropriately prepared from commercial MCC by mixed acid hydrolysis combined with ultrasonication, whether there is an additional swelling treatment or not and whether the form of MCC before mixed acid hydrolysis is powder or slurry. In addition, all the obtained suspensions of spherical CNC exhibited an excellent dispersion stability, as shown in Appendix A; thus, they are convenient for long-term storage without any special protection measures.

### 2.3. Crystallinity

XRD was performed to examine the changes in the crystalline structure and the crystallinity of MCC during the pretreatment. Figure 5a shows the XRD patterns of MCC, MCC+NaOH, and MCC+NaOH+DMSO. In the XRD pattern of MCC, there are diffraction peaks at 2*θ* around 14.9°, 16.3°, 22.5°, and 34.6° that correspond to the (1$\bar{1}$0), (110), (200), and (004) crystallographic planes, in accordance with the characteristic diffraction peaks of cellulose Ⅰ_β_ [16,38,39]. The diffractograms of other samples show peaks located at 2*θ* around 12.0°, 20.0°, and 22.0°, which are characteristic of the (1$\bar{1}$0), (110), and (200) crystallographic planes of cellulose II. The results indicate that the crystal type of pretreated MCC has been transformed from cellulose Ⅰ to cellulose Ⅱ through alkaline treatment, whereas the swelling treatment by DMSO has no effect on the crystal type. These XRD findings are supportive of and consistent with the FTIR results.

The crystallinity of different samples was calculated according to the XRD results as listed in Table 1. Compared with the 78.85% crystallinity of commercial MCC, the crystallinity of MCC+NaOH is lower—only 58.56%—which is attributed to the transformation of crystal type. In Liu’s study, the crystallinity of cellulose also decreased significantly after alkaline treatment [24]. Meanwhile, the crystallinity of MCC+NaOH+DMSO increased to 64.81% because of the removal of residual small molecules by DMSO.

The average cross-sectional dimensions of the elementary crystallites perpendicular to the (1$\bar{1}$0), (110), and (200) crystallographic planes of different samples were quantified using the XRD results and the Scherrer equation as listed in Table 1 [39]. Due to the transformation of crystal type, there was no correlation between the crystallite size data of commercial MCC and pretreated MCC. After further swelling treatment by DMSO, the regularity of the crystallite size data is consistent with that of alkaline treatment only, in which the crystallite size of (110) is the largest, (200) is the second, and (1$\bar{1}$0) is the smallest, but all values are increased. This indicates that the swelling treatment by DMSO contributes to the growth of cellulose unit cells through swelling and drying processes.

The diffractograms in Figure 5b show the characteristic peaks of cellulose II, indicating that the crystal type of pretreated MCC is maintained in all the obtained CNC samples. As there is a crystal transformation during the pretreatment, it is meaningless to compare the crystallinity of commercial MCC and CNC samples. Therefore, compared with the pretreated MCC, the crystallinity of all the obtained CNC samples is obviously increased, as listed in Table 1. This indicates that mixed acid hydrolysis causes a preferential degradation of amorphous regions in the material structure, whereas crystalline regions are domains with a higher resistant ability to the mixed acid [19,21]. Acid hydrolysis treatment increasing the crystallinity of CNC by the removal of disordered structure in MCC or cellulose fibers can be found in previous reports [2,15,21], but some studies still support the opposite conclusion [40,41].

The crystallinity of CNC(N_P_-H) is 78.30%, which is higher than that of CNC(N_S_-H) at 74.11%, meaning that the form of slurry or powder before the mixed acid hydrolysis exerts a certain impact on the crystallinity of these CNC samples. It can be inferred that in slurry form, the mixed acid can enter the pretreated MCC, which provides easier contact with the crystalline region. While the amorphous region is hydrolyzed, a small part of the crystalline region will be destroyed as well. While in powder form, MCC can only be hydrolyzed gradually from the outside, giving priority to the amorphous region; thus, its crystallinity is higher. However, the crystallinity of CNC(N_S_-D_P_-H) and CNC(N_S_-D_S_-H) is nearly the same, meaning that the form before the mixed acid hydrolysis has no impact on the crystallinity of the CNC samples obtained with the pretreatments of alkaline treatment and swelling treatment. This may be due to the good swelling effect of DMSO. In powder form, there are many cracks caused by the incomplete recovery of deformation after swelling by DMSO, providing the mixed acid easy contact with the crystalline region [42]. Therefore, there is basically no difference in their crystallinity.

The crystallite size of CNC samples was also calculated (Table 1). The regularity of the crystallite size data is different from that of the pretreated MCC, in which the crystallite size of (1$\bar{1}$0) is the largest, (1$\bar{1}$0) is the second, and (200) is the smallest, and all values are clearly increased. This phenomenon also exists in other works [2,15,43]. Ahmed-Haras explained that the growth of cellulose crystals may be due to the partial removal of the MCC amorphous region by acid hydrolysis [2]. Although mixed acid may destroy the crystalline region, it is more inclined to hydrolyze the amorphous region, resulting in an increase in crystallinity at the macro level and the rearrangement of cellulose chains or cellulose cell units at the micro level.

### 2.4. Thermal Stability

To evaluate the thermal stability of commercial MCC and pretreated MCC, thermogravimetric analysis was performed as shown in Figure 6a,b. Their thermal degradation processes can be divided into four steps: the loss of absorbed water before 100 °C, the appearance of a plateau until 220 °C, the degradation of cellulose at temperatures ranging from 220 to 400 °C, and the carbonation of the residual products over 400 °C [20].

The thermal degradation parameters such as the onset degradation temperature (T_0_), the maximum degradation temperature (T_Max_), and the corresponding weight loss (WL) are calculated from the TG and DTG curves as listed in Table 2. Compared with commercial MCC, the water absorption capacity of pretreated MCC increases significantly, which is attributed to the full exposure of cellulose [43]. In particular, the hydrogen bond in MCC can be destroyed by hot water, and new hydroxyl groups will be introduced obtaining more adsorbed water. Due to its higher crystallinity, the T_0_ of MCC is higher than that of pretreated MCC, while their T_Max_ is basically the same. The above results show that the thermal stability of MCC decreased slightly after pretreatment, but the hydrophilicity increased.

Figure 6c,d shows the thermogravimetric analysis diagram of CNC samples prepared with different pretreatments. Their thermal degradation processes can be divided into another four steps: the loss of absorbed water before 100 °C, the loss of bonded water and degradation of cellulose at temperatures ranging from 100 to 290 °C (main stage Ⅰ as listed in Table 2), the slow charring process of the solid residue at temperatures ranging from 290 to 400 °C (main stage Ⅱ as listed in Table 2), and the carbonation of the residual products over 400 °C. Previous literature also mentions these two main stages in the thermal decomposition process of CNC [16,33,34,40].

The thermal degradation parameters of CNC samples were calculated from the TG and DTG curves which are also listed in Table 2. Compared with the T_0_ (at around 300 °C) of pretreated MCC, the T_0_ of CNC samples is sharply decreased to about 200 °C. The T_Max_ of CNC samples in main stage I is much lower than the T_Max_ of pretreated MCC, while the T_Max_ of CNC samples in main stage II is slightly higher than that of pretreated MCC. The obvious decrease in the thermal stability of CNC samples is because the sulfate groups introduced on cellulose chains can catalyze the thermal degradation of cellulose [20,34,41]. Furthermore, the high surface area of CNC samples may also play an important role in diminishing their thermal stability due to the increased exposure of the surface area to heat [1,16]. Moreover, the decomposition of CNC samples occurring at lower temperatures may also indicate a faster heat transfer in CNC samples [34,38]. It also can be seen that the amount of char residue in CNC samples is noticeably larger than that of pretreated MCC [34]. Wang et al. explained that CNC particles had a great number of free end chains due to their small particle size, which started decomposition at a lower temperature facilitating the increase in the char yield of CNC [40].

In comparison with the thermal degradation parameters of CNC samples, CNC samples with different thermal stability can be obtained from different pretreatments. The initial weight loss of CNC(N_S_-H) is higher than other CNC samples because of more absorbed water and bonded water. At the same time, more bonded water will lead to a significant decrease in its T_0_, which is 198.1 °C. Due to its higher crystallinity, the T_0_ of CNC(N_P_-H) is 236.3 °C. Meanwhile, the T_0_ of CNC(N_P_-D_S_-H) is lower than that of CNC(N_P_-D_P_-H): 206.3 and 220.5 °C, respectively. A reasonable explanation is that the amorphous regions are more concentrated in CNC(N_P_-D_P_-H) than in CNC(N_P_-D_S_-H), which is more readily thermally decomposed. In addition, the T_Max_ values of CNC samples in main stage Ⅰ show no significant differences, ranging from 265.8 to 271.2 °C. Interestingly, the T_Max_ and WL of all CNC samples in main stage Ⅱ are basically the same, meaning that the slow charring process of the solid residue at temperatures ranging from 290 to 400 °C has nothing to do with the pretreatment of MCC.

From the above results, it can be concluded that the pretreatment of MCC has a certain influence on the properties of the spherical CNC subsequently obtained by mixed acid hydrolysis. Table 3 shows the pretreatment and hydrolysis processes of preparing spherical CNC by mixed acid hydrolysis in existing works. Among them, cellulose fiber and MCC were usually pretreated in the same way, and the rationality of this was not systematically investigated. Meanwhile, spherical CNC can also be obtained from MCC by acid hydrolysis without any pretreatment, from which can be drawn the same argument as this work—that DMSO swelling is not indispensable for the preparation of spherical CNC by acid hydrolysis. In addition, the mixed acid hydrolysis time of 2 h is sufficient, which can reduce carbonization and energy consumption.

## 3. Materials and Methods

### 3.1. Materials

MCC with a particle size of 25 μm and sodium hydroxide (NaOH, 96%) were purchased from Macklin Biochemical Co, Ltd. (Shanghai, China). Dimethyl sulfoxide (DMSO, ≥98%) was supplied by Aladdin Biochemical Technology Co., Ltd. (Shanghai, China). Sulfuric acid (H_2_SO_4_, 95–98%) and hydrochloric acid (HCl, 36–38%) were bought from Shuanglin Chemical Reagent Co., Ltd. (Hangzhou, China). All chemicals were used as received.

### 3.2. Pretreatment of MCC

#### 3.2.1. Alkaline Treatment of MCC

MCC (30 g) was treated with a 5 M NaOH (250 mL) aqueous solution under mechanical stirring at 80 °C for 3 h, then filtered and washed with distilled water until pH ˂ 8. The slurry form was obtained at a vacuum of 0.06 MPa. Meanwhile, the powder form was obtained after drying at 40 °C for 24 h.

#### 3.2.2. Swelling Treatment of Alkaline Treated MCC

The obtained powder of alkaline-treated MCC was immersed into DMSO (250 mL) under mechanical stirring at 80 °C for 3 h, then filtered and washed with distilled water. The slurry form after swelling treatment by DMSO was obtained at a vacuum of 0.06 MPa. Meanwhile, the powder form was obtained after drying at 40 °C for 24 h. The pretreatment routes of MCC are shown by a flow diagram in Figure 7.

### 3.3. Mixed Acid Hydrolysis

One quarter of pretreated MCC (slurry form or powder form) was hydrolyzed with mixed acid (250 mL) at 80 °C under mechanical stirring and ultrasonication (40 KHz, 100 W, KQ2200DV, Shumei, China). The mixed acid was prepared by mixing sulfuric acid, hydrochloric acid, and distilled water at a ratio of 3:1:6 (*v*/*v*). Due to the ultrasonication during spherical CNC preparation, the acid molecules quickly penetrated into the inner amorphous region of the cellulose fibrils. Therefore, the hydrolysis reaction simultaneously occurred at the surface and in the inner amorphous regions of MCC. This caused MCC to first hydrolyze to the sub-micrometer fragments, instead of directly obtaining the CNC. Moreover, because the mixed acid concentration was mild, the dissolving capability of the cellulose fragments was very limited. Thus, the sub-micrometer fragments were further hydrolyzed by the mixed acid to form spherical CNC particles gradually [11,12,39,42].

The hydrolysis time in this work was fixed at 2 h to avoid the carbonization caused by further hydrolysis as shown in Appendix A. After hydrolysis, the suspensions were immediately diluted using an 8 times volume of distilled water to terminate the hydrolysis reaction. Then, the suspensions were centrifuged at 10,000 rpm for 10 min to remove the excessive acid solution and washed with distilled water several times. Finally, the washed CNC was dialyzed using dialysis membranes (MWCO 8000–14,000, 44 mm) against distilled water for three days until the pH was constant. Samples prepared through different routes are recorded as CNC(N_P_-D_P_-H), CNC(N_P_-D_S_-H), (CNC(N_P_-H) and CNC(N_S_-H), respectively. Among them, the route of preparing CNC(N_P_-D_S_-H) is the common method in existing works as marked with red lines.

### 3.4. Characterization

#### 3.4.1. Fourier Transform Infrared (FTIR) Spectroscopy

MCC or CNC samples were mixed with KBr to produce tablets. FTIR spectra were conducted on a Nicolet 5700 spectrophotometer (MI, USA) in the range of 4000 to 400 cm^−1^ at a resolution of 2 cm^−1^ over 30 scans.

#### 3.4.2. Morphological Investigation

The morphology of MCC and CNC samples was observed using a field-emission scanning electron microscope (FE-SEM, FEI Quanta FEG 650, Hillsboro, OR, USA). MCC samples were directly laid on the conductive adhesive. The CNC suspensions were diluted to 0.01 wt.% with distilled water, and then dropped on the conductive adhesive and dried at room temperature. Then, these samples were coated with platinum to make them conductive prior to the analysis. The ImageJ software was used to calculate the particle size of CNC samples from the FE-SEM images.

#### 3.4.3. X-ray Diffraction (XRD) Analysis

The XRD spectra of MCC and CNC samples were measured with the X-ray diffractometer (D8 Advance, Bruker, Germany) using Cu-Kα radiation (λ = 0.15418 nm) at an accelerating voltage of 40 kV and a current of 40 mA.

The data were collected in the range of 2*θ* = 5° to 45°. The degree of crystallinity could be relatively expressed by the percentage crystallinity index (CrI, %). The equation used to calculate the CrI was described by Segal et al. [44] in the following form,
(1)$$\mathrm{CrI}\;\left(\%\right)=\left[\left(I_{200}-I_{\mathrm{am}}\right)/I_{200}\right]\times 100$$
where, $I_{200}$ is the counter reading at peak intensity at a 2*θ* angle close to 22° representing the crystalline part and $I_{\mathrm{am}}$ is the counter reading at peak intensity at 2*θ* = 18° representing the amorphous part in cellulose.

The crystallite size was calculated using the Scherrer equation,
(2)D=Kλ/βcosθ
where, D is the “apparent crystallite size”, β is the full width of the diffraction peak measured at half maximum height (FWHM) by Lorentz function, and the constant K is equal to 0.94 [45].

#### 3.4.4. Thermal Analysis

Thermogravimetry (TG) and derivative thermogravimetry (DTG) curves of MCC and CNC samples were obtained using a Netzsch TG209 F3 instrument (Bavaria, Germany) over a temperature range of 30 °C to 800 °C at a heating rate of 10 °C/min under nitrogen supply.

## 4. Conclusions

In this study, spherical CNC samples with an average size of around 10 nm were successfully prepared by mixed acid hydrolysis from commercial MCC with different pretreatment routes. The physicochemical properties of spherical CNC were evaluated by FT-IR, FE-SEM, XRD, and TG. The characterization results revealed that an additional swelling process is not indispensable for the preparation of spherical CNC from MCC by mixed acid hydrolysis. Spherical CNC can also be readily obtained from alkaline-treated MCC without any other pretreatment. Furthermore, the form of pretreated MCC before acid hydrolysis (slurry or powder form) also has a certain influence on the morphology, crystallinity, and thermal stability of spherical CNC. This comparative study provides support for further studies in selecting pretreatment processes of MCC to economically prepare desired spherical CNC with a high repeatability.

## Figures and Tables

**Figure 1 ijms-23-10764-f001:**
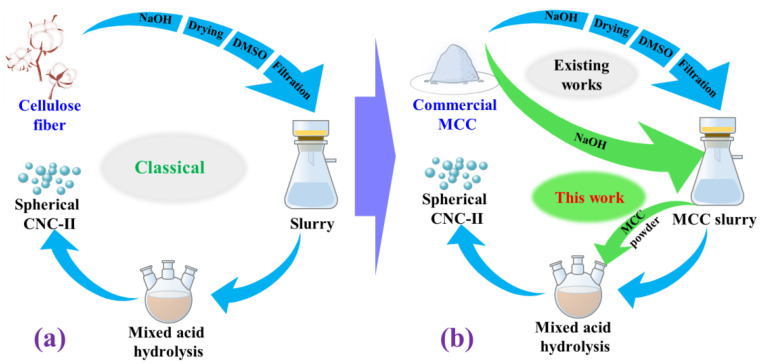
Schematic process flow of (**a**) spherical cellulose nanocrystal (CNC) preparation from cellulose fiber, and (**b**) spherical CNC preparation from commercial microcrystalline cellulose (MCC).

**Figure 2 ijms-23-10764-f002:**
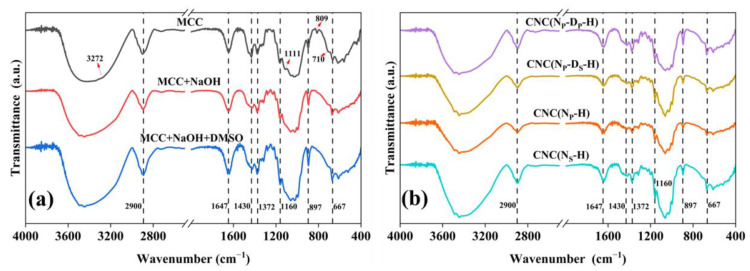
(**a**) FTIR spectra of MCC, MCC after alkaline treatment, and MCC after alkaline treatment and swelling treatment; (**b**) FTIR spectra of CNC samples prepared through different pretreatment routes.

**Figure 3 ijms-23-10764-f003:**
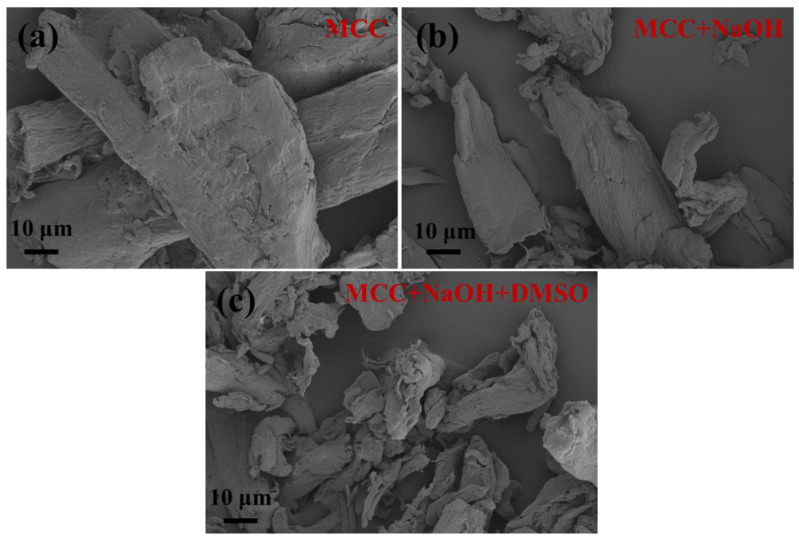
FE-SEM images of (**a**) MCC, (**b**) MCC after alkaline treatment, and (**c**) MCC after alkaline treatment and swelling treatment (1000× magnification).

**Figure 4 ijms-23-10764-f004:**
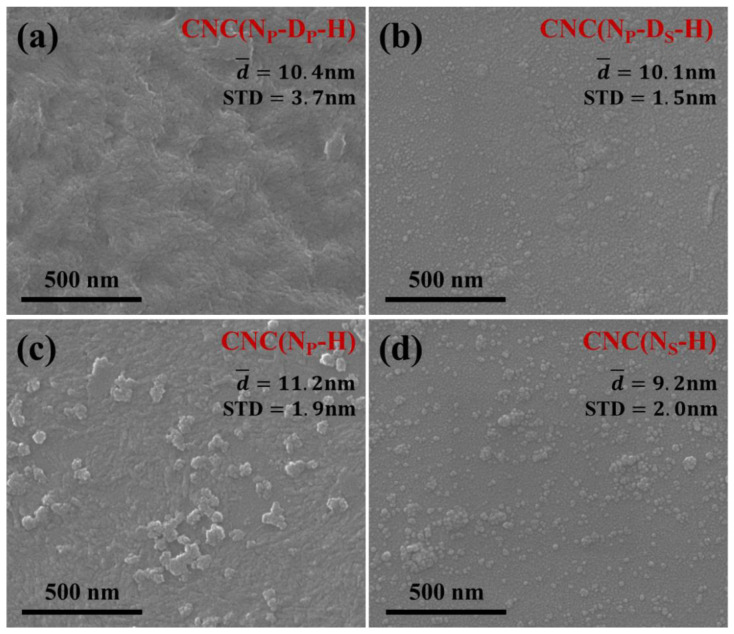
FE-SEM images of CNC samples prepared through different pretreatment routes (200,000× magnification).

**Figure 5 ijms-23-10764-f005:**
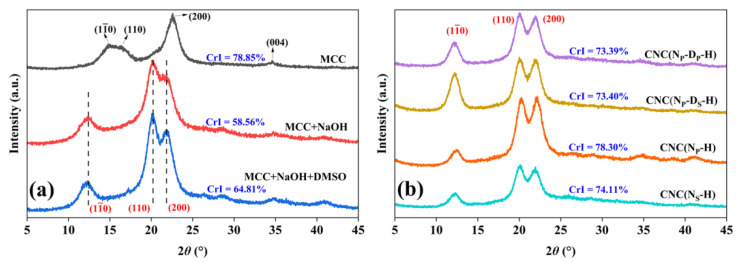
(**a**) XRD patterns of MCC, MCC after alkaline treatment and MCC after alkaline treatment and swelling treatment; (**b**) XRD patterns of CNC samples prepared through different pretreatment routes.

**Figure 6 ijms-23-10764-f006:**
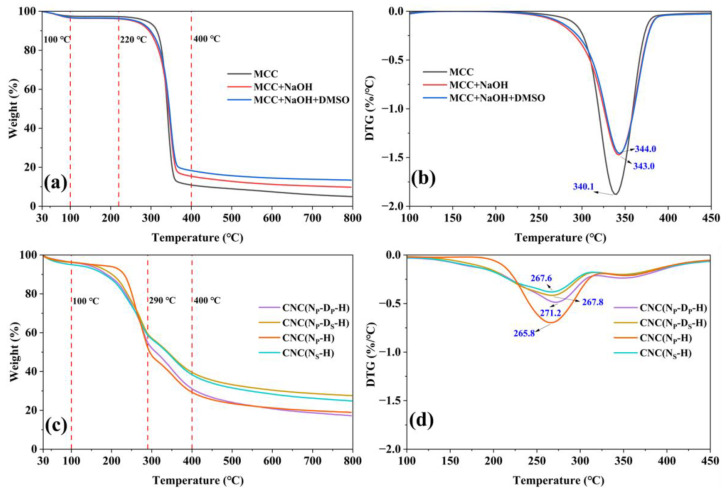
(**a**) TG and (**b**) DTG curves of MCC, MCC after alkaline treatment and MCC after alkaline treatment and swelling treatment; (**c**) TG and (**d**) DTG curves of CNC samples prepared through different pretreatment routes.

**Figure 7 ijms-23-10764-f007:**
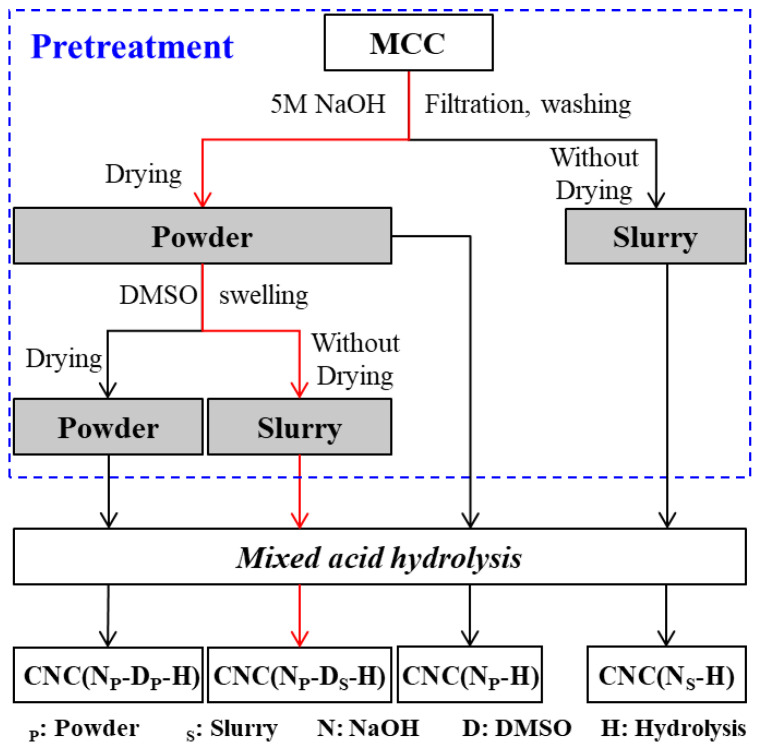
Pretreatment routes of MCC and its products after mixed acid hydrolysis (the common method in existing works is marked with red lines).

**Table 1 ijms-23-10764-t001:** The crystallinity and crystallite size for different samples.

Sample	CrI (%)	Crystallite Size (D, nm)		
		$\left(1\bar{1}0\right)$	(110)	(200)
MCC	78.85	4.06	3.81	5.06
MCC+NaOH	58.56	3.81	4.52	4.08
MCC+NaOH+DMSO	64.81	4.36	5.18	4.65
CNC(N_P_-D_P_-H)	73.39	7.15	5.98	5.87
CNC(N_P_-D_S_-H)	73.40	7.42	6.13	5.65
CNC(N_P_-H)	78.30	7.51	6.10	5.47
CNC(N_S_-H)	74.11	7.24	5.99	5.69

**Table 2 ijms-23-10764-t002:** Thermal degradation parameters of different samples.

Sample	Initial Weight Loss (%)	T_0_ (°C)	Main Stage I		Main Stage II		Final Residue (%)
			T_Max_ (°C)	WL (%)	T_Max_ (°C)	WL (%)	
MCC	2.45	315.1	340.1	84.62	–	–	4.97
MCC+NaOH	3.34	295.5	343.0	78.22	–	–	9.79
MCC+NaOH+DMSO	3.28	299.0	344.0	75.60	–	–	13.41
CNC(N_P_-D_P_-H)	3.70	220.5	271.2	42.56	348.7	22.94	16.72
CNC(N_P_-D_S_-H)	3.67	206.3	267.8	36.96	347.8	19.82	27.60
CNC(N_P_-H)	3.77	236.3	265.8	48.18	353.3	18.81	18.97
CNC(N_S_-H)	4.95	198.1	267.6	36.37	350.1	20.29	24.86

**Table 3 ijms-23-10764-t003:** Spherical CNC (cellulose Ⅱ) prepared by mixed acid hydrolysis with different pretreatments.

Raw Material	Pretreatment	Form before Hydrolysis	Mixed Acid Hydrolysis	CrI (%)	T_0_ (°C)	Particle Size (nm)	Ref.
Cotton fiber	NaOH, 75 °C, 4 h; DMSO, 75 °C, 4 h	Slurry	H_2_SO_4_:HCl = 3:1; 75 °C, Ultrasonic, 8 h	-	-	50	[12]
Cellulose fiber	NaOH, 80 °C, 3 h; DMSO, 80 °C, 3 h	Slurry	H_2_SO_4_:HCl = 3:1; 80 °C, Ultrasonic, 8 h	82.0	-	80	[3]
Cellulose fiber	NaOH, 80 °C, 3 h; DMSO, 80 °C, 3 h	Slurry	H_2_SO_4_:HCl = 3:1; 80 °C, Ultrasonic, 8 h	-	-	5.9–10.9	[25]
MCC	NaOH, 80 °C, 3 h; DMSO, 80 °C, 3 h	Slurry	H_2_SO_4_:HCl = 3:1; 80 °C, Ultrasonic, 8 h	81.3	-	60	[4]
MCC	NaOH, 80 °C, 3 h; DMSO, 80 °C, 3 h	Powder	H_2_SO_4_:HCl = 3:1; 80 °C, Ultrasonic, 2 h	73.4	220.5	10.4	This work
MCC	NaOH, 80 °C, 3 h; DMSO, 80 °C, 3 h	Slurry	H_2_SO_4_:HCl = 3:1; 80 °C, Ultrasonic, 2 h	73.4	206.3	10.1	This work
MCC	NaOH, 80 °C, 3 h	Powder	H_2_SO_4_:HCl = 3:1; 80 °C, Ultrasonic, 2 h	78.3	236.3	11.2	This work
MCC	NaOH, 80 °C, 3 h	Slurry	H_2_SO_4_:HCl = 3:1; 80 °C, Ultrasonic, 2 h	74.1	198.1	9.2	This work

## Data Availability

Not applicable.

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
