# Peer review of "Preparation of Spherical Cellulose Nanocrystals from Microcrystalline Cellulose by Mixed Acid Hydrolysis with Different Pretreatment Routes"

_ijms, 2022, doi:10.3390/ijms231810764_

Round 1

Reviewer 1 Report

Line 33 – Change “larger” to “larger”

Line 40 -  The morphology of cellulose nanomaterials is relation with the characteristics and brings about completely different application prospects. (This sentence does not read well. Please rewrite).

 Line 57 – Change “researches” to “studies”

 Figure 1 – It is suggested that figures stand alone. Consider defining your acronyms in the figure legend. For example, insert “microcrystalline cellulose (MCC)”

 Line 88 – Change “researches” to “studies” This change should be made in other places throughout the manuscript.

Suggest that the authors move the “Materials and Methods” section 3  to section 2 and move the results section 2 to section 3.

Reviewer 2 Report

In my opinion, the authors of this article are sumarising weel the study they performed in Abstract of the manucript. The introduction provides sufficient details regarding previous studies in the field. It contains 31 references and a fgure which could help the readers to better understand the purpose of the study. Results and discussion part is is clearly structured. For each individual analysis, sufficient explanations and comparisons with the data obtained in other studies are provided. The total number of references is 45, wich in my opinin these are not too many or too few. The results obtained from different experiments are correlated with each other. Possible explanations for the differences obtained between the analyzed samples are provided. Figures are at good resolution and the figure captions are clear. In the experimental part, there are enogh deteils to allow the experiments to be reproduced. Conclusions are short and concise and are appropriate. They are suported by the results.

 I have a sugestion:

- in figure 2 - FTIR spectra, on y axis is written "Transparent"; I did not see any FTIR spectra in literature with "Transparent" on y axis; usualy is Transmitance T% or Absorbance

-

Reviewer 3 Report

- Usually, infrared spectra are plotted x (wavenumber cm-1) and y (transmittance a.u.) axis.

- All samples MCC, MCC after alkaline treatment, MCC after alkaline treatment and swelling treatment, and CNC prepared by different pre-treatment routes did not show the presence or decrease of lignin and hemicellulose groups in their chemical compositions. It is necessary to address this discussion in the FTIR

- Suggestion to move figure 3 to the topic “2.2 Micromorphology” after introducing the technique.

- The authors indicate in lines 158-159 that spherical CNC can be well-prepared from commercial MCC by mixed acid hydrolysis combined with ultrasound. However, the discussion should be improved. What are the main contributions that ultrasound has promoted to CNC? The literature reports that ultrasound waves generate more defibrillation and separation of nanofibers and, consequently, a reduction in fiber diameters.

- Line 161-163 "all obtained spherical CNC suspensions exhibit excellent dispersion stability as shown in Fig. S1” this discussion should be complemented with another analysis, zeta potential suggestion, in which it will investigate the electrostatic stability of the samples produced.

- Suggestion to move figure 6 to the topic “2.4 Thermal stability” after introducing the technique.
